# A Simple Method for the Synthesis of a Coral-like Boron Nitride Micro-/Nanostructure Catalyzed by Fe

**DOI:** 10.3390/nano13040753

**Published:** 2023-02-17

**Authors:** Yanjiao Li, Xueren Wang, Jian Wang, Xinfeng Wang, Dejun Zeng

**Affiliations:** 1Foundation Department, Rocket Force University of Engineering, Hongqing Town, Xi’an 710025, China; 2Zhijian Lab, Rocket Force University of Engineering, Hongqing Town, Xi’an 710025, China; 3Beijing System Engineering Institute, Beijing 100101, China; 4School of Materials Science and Engineering, Chang’an University, Xi’an 710064, China

**Keywords:** boron nitride, coral-like, micro-/nanostructure, ball milling and annealing, mechanism

## Abstract

Catalyzed by Fe, novel a coral-like boron nitride (BN) micro-/nanostructure was synthesized from B_2_O_3_ by a ball milling and annealing process. Observations of the morphology of the product indicated that the coral-like BN micro-/nanostructure consists of a bamboo-shaped nanotube stem and dense h-BN nanoflakes growing outward on the surface of the nanotube. Experimental results showed that the morphology of the BN nanotube was greatly dependent on the anneal process parameters. With the annealing time increasing from 0.5 h to 4 h, the morphology developed from smooth BN nanotubes, with a diameter size of around 100 nm, to rough, coral-like boron nitride with a large diameter of 3.6 μm. The formation mechanism of this coral-like BN micro-/nanostructure is a two-stage growth process: bamboo-shaped BN nanotubes are first generated through a vapor–liquid–solid (VLS) mechanism and then nanoflakes grow surrounding the surface of the nanotube. Acid pickling and a hydrolysis process were carried out to remove Fe, iron nitrogen and unreacted B_2_O_3_ impurities.

## 1. Introduction

Since the discovery of carbon nanotubes (CNTs) by Iijima in 1991, they have become a research hotspot due to their unique structure and excellent mechanical, thermal and electrical properties [1,2]. Based on the expectation of their special properties, boron nitride nanotubes (BNNTs) with a similar structure to carbon nanotubes have gradually attracted the attention of researchers, and related research studies have been carried out. In 1994, Rubio et al. [3] predicted the stable existence of boron nitride nanotubes by tightly bound molecular dynamics, and in 1995, Chopra et al. [4] successfully prepared BNNTs by the plasma arc discharge method, which initiated the research on BNNTs. Over the past two decades, researchers have extensively explored the preparation, purification, performance and application of BNNTs. Although industrial production has not yet been achieved, research on the preparation technology is developing rapidly. Various synthesis methods, such as arc discharge [5,6,7], laser ablation [8,9,10], chemical vapor deposition (CVD) [11,12,13,14], the ball-milling method [15,16,17,18], the thermal plasma jet method [19,20,21,22,23] and others [24,25,26,27], have been proposed and investigated. After more than 20 years of research, the yield of nanotubes has increased from the milligrams scale to the gram scale. In 2014, a group at the National Research Council Canada (NRC) successfully increased the yield to 20 g/h by an RF induction plasma jet process [21]. In the same year, Zettle et al. increased the yield to 35 g/h by the extended pressure inductively coupled plasma system (EPIC) method [23]. Based on the research of preparation technology, researchers have also extensively explored the properties of BNNTs and found that they have excellent mechanical properties [28,29], good thermal conductivity [30,31], outstanding oxidation resistance [32,33], novel photoluminescence properties [34,35,36], unique magnetic properties [37] and stable superhydrophobic properties [38,39,40,41], in addition to their low density, insulation, non-toxicity and good biocompatibility [42,43]. Due to these excellent properties, boron nitride nanotubes can be used in oil filtering, self-cleaning membranes, hydrogen storage, heat dissipation, drug carriers, optoelectronics, etc. [44,45,46].

It is well known that the physicochemical properties of inorganic materials are closely related to their structure and morphology. Different structural topography can lead to different properties and, therefore, different application areas. Similarly, based on the expectation of their excellent properties, researchers have gradually become interested in boron nitride nanomaterials with different morphologies. Nowadays, various types of BN nanostructures, such as BN nanoparticles [47,48,49], BN nanosheets [50,51,52], BN nanocapsules [53], BN nanoropes [54], BN nanoribbons [55], BN aerogels [56,57], BN single domains [58], BN films [59] and hexagonal boron nitride-based heterostructures [60], have attracted worldwide interest, not only for their fundamental scientific interest, but also for their unique properties and potential applications. Besides the BN nanostructures mentioned above, a new type of boron nitride structure with nanoflakes, whiskers, burrs or filaments growing on the surface of nanotubes was also reported, which has been named according to its different groups as a thorn-like BN nanostructure [61], collapsed BN nanotubes [62], surface-modified BN nanotubes [63,64], nanotubes decorated with BN nanosheets [65], a coral-like BN nanostructure [66], a BN micro-/nanostructure [67], a boron nitride hierarchical structure [68,69] or a boron nitride nanosheets–nanotubes hybrid structure [70]. Due to its coral-like shape and large number of nano-sized flakes or whiskers growing on its micron-sized surface, the authors of this paper refer to it as a coral-like boron nitride micro-/nanostructure. At present, there are few reports on this novel boron nitride structure. Existing reports have mainly focused on the preparation method and growth mechanism, and only several papers have focused on performance research. The preparation methods reported in the literature are relatively complex [66,68], and the raw materials used by some methods are toxic [67]. The preparation of this BN micro-/nanostructure using a simple method and safe raw materials is a problem worthy of study. The ball milling and annealing method is a simple method for synthesizing boron nitride nanotubes, which was first reported by Chen [15]. Many research groups have applied or improved this method to prepare boron nitride nanotubes [16,17,18]; however, there is no report on the use of this method to prepare coral-like boron nitride micro-/nanostructures.

In this paper, catalyzed by Fe, a coral-like boron nitride micro-/nanostructure prepared by a ball milling and annealing method, using cheap and safe B_2_O_3_ as the raw material, is reported. The formation mechanism, purification technology and potential applications of this novel boron nitride micro-/nanostructure is also discussed. 

## 2. Materials and Methods

B_2_O_3_ powder (99.99% purity) and micron-sized Fe powder (99.99% purity) in a molar ratio of 1:1 were first milled together in a planetary ball mill from Tianchuang instrument and equipment Co., Ltd (Changsha, China). The weight ratio of the milling balls to B_2_O_3_ powder was 100:1. The grinding tank and the grinding medium were both made of stainless steel. The rotation speed of the ball mill was 280 rpm and the ball milling process lasted for 4 h under the protection of high-purity nitrogen. The milled boron oxide/iron mixed powder was then annealed in a GLS-1600X tube furnace manufactured by Kejing (HeFei, China) at 1200 °C for 2 h under a high-purity ammonia gas flow at 60 mL/min. The B_2_O_3_ powder was provided by Mountain technology development center for non-ferrous metals (Beijing, China). The Fe powder was provided by Bowei applied materials technology Co., Ltd. (Shanghai, China). The ammonia gas was provided by Tenglong Chemical Co., Ltd (Xian, China). 

The crystalline structure of the product was investigated by Bruker D8 Venture X-ray diffraction analysis (XRD) (Salbruken, Germany) using Cu Ka radiation (λ = 0.15418 nm) at room temperature. The Raman spectroscopy measurement was performed by a Hrobia Xplra Plus Raman spectrometer manufactured by Horiba Scientific Co., (Paris, France), λ = 532 nm. S-4800 scanning electron microscopy (SEM) provided by Hitachi (Tokyo, Japan) and JEM-200CX transmission electron microscopy (TEM) provided by JEOL (Takashima, Tokyo, Japan) were employed to observe the morphology of the product. Selected area electron diffraction (SAED) of the sample was performed via TEM. JEM-2100F high-resolution transmission electron microscopy (HRTEM) provided by JEOL (Takashima, Tokyo, Japan) was employed to observe the high-power and high-resolution photos of the products. X-ray energy dispersive spectroscopy (EDS) attached to high-resolution transmission electron microscopy was used to analyze the chemical composition of the sample. The impurity content of the sample was measured by a Bruker S8 Tiger X-ray fluorescence spectrometer (Salbruken, Germany). 

## 3. Results

### 3.1. Synthesis and Characterization

Figure 1 shows the SEM images and XRD patterns of the raw materials. Figure 1a demonstrates that the morphology of the B_2_O_3_ powder is irregular particles with a size in the range from 50 μm to 500 μm. Figure 1b gives the XRD pattern of the B_2_O_3_ powder shown in Figure 1a. Strong B_2_O_3_ peaks indicated that the B_2_O_3_ powder is crystalline. Figure 1c is the SEM image of the original Fe powder. It can be observed that the morphology of the Fe particles is spherical and their diameter is in the range from 0.5 μm to 3 μm. The XRD pattern shown in Figure 1d indicates that the Fe powder was in a crystal state. 

After being milled for 4 h in high-purity N_2_, the color of B_2_O_3_ powder doped with Fe powder changed from white to black. Figure 2a shows the SEM image of the mixed B_2_O_3_/Fe powder. It can be observed that the B_2_O_3_ powder transformed into uniform equiaxial particles with a size of about 3 μm to 10 μm, which is obviously smaller than the original B_2_O_3_ particles. The particles become rough and loose. Some spherical Fe particles can also be observed. Figure 2b shows the XRD patterns of the mixed B_2_O_3_/Fe powder after being milled for 4 h. Weak crystal peaks of B_2_O_3_ can be observed in the XRD spectrum, which indicates that most of the B_2_O_3_ changed into the amorphous state after ball milling and only a small amount of crystalline B_2_O_3_ exists. The strong crystal peaks of Fe indicate that Fe remained in its crystalline state. 

The optical image, XRD pattern and Raman spectra of the product are shown in Figure 3a–c. The product obtained after annealing was a white loose powder with small black substances, as shown in Figure 3a. The size of the porcelain boat is 60 mm × 30 mm × 15 mm. The XRD pattern of the product indicates that it was mainly composed of H-BN. Weak Fe, Fe_2_N and Fe_4_N peaks showed that the product contained a small amount of these impurities. Raman spectra exhibited specific peaks at 1378 cm^−1^, corresponding to the H-BN vibration mode (E_2g_).

To identify the morphology of the product, SEM of the product with different magnifications was undertaken, as shown in Figure 4a–d. Some clusters composed of fiber-like matter can be observed in Figure 4a. The diameter of a single cluster is about 50 μm~100 μm. Figure 4b shows the SEM image of a single cluster. It can be observed that the cluster was constituted by a lot of uniform fibrous substances intertwined together. The surface of the fibrous substances is rough and their diameter is uniform. An enlarged SEM image of the cluster shown in Figure 4b indicates that many petal-like flakes grow outward vertically on the surface of the trunk, which is why it appears rough, as shown in Figure 4c. To observe these flakes carefully, a enlarged SEM image of the single fibrous substances indicated by an arrow in Figure 4c is provided, as shown in Figure 4d. It can be clearly observed that the flakes resemble a petal; their edges are slightly curved and the thickness of the curve edge is about 10 nm, so it can be speculated that the actual thickness of the flakes is about 2~5 nm, which is very thin. The area of these flakes is essentially the same, being approx. 200 nm~300 nm in width and 200 nm~300 nm in length. Since these BN structures resemble a coral, with a micron-sized external diameter and nano-scale flakes growing outward on the surface of the main stem, we named it as a coral-like BN micro-/nanostructure. 

Figure 5a is the TEM image of the coral-like BN micro-/nanostructure. It can be found that numerous hairy flakes grow outward on the surface of the main stem and its trunk is a bamboo-shaped tubular structure. Due to the shielding of the thick flakes on the surface, the bamboo-shaped tubular structure of some tubular structures cannot be observed clearly, as shown by arrows. Figure 5b is the EDS spectrum of the BN micro-/nanostructures shown in Figure 5a. The atomic ratio of B:N is 33.46:36.22 based on EDS analysis, which is in good agreement with the chemical stoichiometric ratio of BN. The signals of C and Cu arise from the preparation of the TEM sample and the signals of O originate from impurities. Hence, it can be concluded that the coral-like micro-/nanostructures shown in Figure 5a are BN micro-/nanostructures. Figure 5c shows the HRTEM images of the flakes, where highly orderly and clear lattice fringes can be observed. The inter-layer spacing between adjacent fringes is about 0.334 nm, corresponding to the (002) planes of h-BN. The lattice fringes with different directions interlaced together indicate that several layers of flaky structures are stacked together. Single direction lattice fringes on the right show this region as a single-layer structure, as shown by the arrow. This monolayer appears to be transparent, which indicates that its thickness is very thin. The SAED pattern identified that the flakes were a hexagonal BN crystal structure.

### 3.2. Formation Mechanism

In order to investigate the formation mechanism of the BN micro-/nanostructure, the effect of annealing time on the morphology of the product was investigated while keeping other parameters unchanged, as shown in Figure 6. Very smooth fibrous matter with a small diameter can be observed when the annealing time was 0.5 h. With the increase in the annealing time, as shown in Figure 6b, the diameter of the sample becomes large and some fragments appear on their surface. When the annealing time increases to 2 h, thick petal-like sheets grow outward on the surface of the sample and their morphology turns coral-like. With a continued increase in the annealing time, as shown in Figure 6d, the morphology of the sample is almost unchanged compared with that annealed for 2 h, except for the obvious increment in the diameter. The above phenomenon indicates that the formation process of the coral-like BN micro-/nanostructure is a two-stage process. Bamboo-shaped BN nanotubes with a smooth surface are generated first, followed by many nanosheets gradually growing outwards on the surface of these BN nanotubes. A schematic illustration of the specific growth process is shown in Figure 7. The reactions involved in the formation of this coral-like BN micro-/nanostructure most likely occur as follows: NH_3_(g) → N_2_(g) + H_2_(g)(1)
B_2_O_3_(s) + 3H_2_ (g) → 2B(s) + 3H_2_O(g)(2)
2B(g) + N_2_(g) → 2BN(s)(3)

Firstly, ammonia is gradually decomposed into hydrogen and nitrogen during the heating process, as shown in reaction formula (1). With the further increase in temperature, hydrogen will react with activated boron oxide to generate boron and water, as shown in formula (2). Compared with the initial state, the Fe particle catalyst is activated after ball milling, so its melting point decreases. When the temperature reaches 1150 °C~1200 °C, the activated Fe particles begin to melt slowly into small droplets, and continuously absorb the boron and nitrogen atoms in the reaction environment, forming small alloy droplets containing Fe-B-N, as shown in Figure 7a. Boron and nitrogen atoms are absorbed and dissolved continuously, which causes the concentration of the alloy to increase further. When the supersaturated state is reached, BN crystals begin to precipitate from the surface of the droplets in the forms of layer, as shown in Figure 7b. With the continuous supply of raw materials, the BN crystal layer gradually thickens, and the newly formed inner BN crystal has a smaller curvature radius than the outer one, which leads to the increase in the stress between the BN crystal and the catalyst droplet. When the stress accumulates to a certain extent, the small droplets will extrude from the weak defects of the crystal layer. This empty crystal layer forms a bamboo knot of the nanotube. The squeezed droplets continue to absorb the B and N atoms and, in doing so, grow periodically. Many bamboo knots grow together in order to form bamboo-like boron nitride nanotubes, as shown in Figure 7c,d. Since the growth time is so short, the surface of the bamboo-shaped BN nanotubes is usually rough and many defects still exist [71,72]. With the continuous supply of reaction gas, B and N atoms nucleate twice at these defects under residual Fe catalysis. Thus, petal-shaped BN nanosheets grow on the surface of the bamboo-shaped BN nanotubes. A large number of nanosheets growing outwards on the surface of the BN nanotubes result in the formation of the BN micro-/nanostructure, as shown in Figure 7e. 

The SEM image of the BN micro-/nanostructure in Figure 8 seems to justify the above analysis. The head of the single BN micro-/nanostructure marked by an arrow in Figure 8a is shown by the enlarged SEM image in Figure 8b. It can be clearly observed that dense nanosheets grow vertically on the surface of the smooth nanotube. Several vertically grown nanosheets are also observed on the internal surface of the nanotubes. This is further evidence that these nanosheets were formed by secondary growth. Due to the sufficient supply of reaction raw materials around the outer surface, the dense nanosheets formed by secondary growth. However, due to the difficulty of contact between the inner surface and the reaction raw material, nanosheets rarely formed by secondary growth on the inner surface.

### 3.3. Purification Technology

The introduction of impurities during the formation of nanomaterials by the ball milling and annealing method is inevitable. On the one hand, they originate from iron base alloy debris produced by the friction and collision between the stainless-steel ball milling media during the ball milling process. On the other hand, some by-products and unreacted raw materials are also produced during high-temperature annealing. According to the XRD analysis and fluorescence spectrum analysis data of the product, Fe, O and iron nitrogen compounds were the main impurities in the product. Moreover, a small amount of metal elements also exists, such as Al, Cr, Si, and Mn. Taking advantage of the fact that metals easily react with acid to form salt solution, these impurities can be removed by acid pickling. Unreacted boron dioxide can be removed by hydrolysis. 

The operation steps were as follows:

(1) The product was taken out from the porcelain boat and added to the beaker containing deionized water. It was stirred for 30 min with a magnetic stirrer to evenly separate the components in the product. 

(2) Then, the mixed solution was heated in a hot plate to fully hydrolyze the unreacted boron oxide in the product.

(3) An appropriate amount of hydrochloric acid with a concentration of 20% was added into the mixed solution to dissolve iron and other metal elements. After stirring for 30 min, the mixture was transferred from the beaker to a conical flask and allowed to stand for 8 h.

(4) After standing, the liquid in the conical flask turned yellow-green. This yellow-green liquid was judged to be ferric chloride, based on the color. After filtering out the upper liquid, deionized water was used to repeatedly wash the powder for 5~6 times to ensure that the impurities were thoroughly washed.

(5) The washed white powder was then dried in a drying oven and collected in a centrifugal tube. This white powder was the purified coral-like boron nitride micro-/nanopowder.

Figure 9a is the XRD pattern of the product after purification; the inset is the optical image of the product after purification, which shows that the product after purification was a white powder without any black substances. The XRD pattern of the product after acid cleaning shows that only weak peaks of Fe_16_N_2_ still exist, except for the strong crystal diffraction peak of H-BN, i.e., the main component in the product is the H-BN phase, and only a small amount of Fe_16_N_2_ impurity exists. This result proved that the hydrolysis reaction and acid cleaning process can effectively remove the impurities introduced in the process of ball milling and annealing, to ultimately obtain BN products with high purity. Figure 9b shows the change of the major impurity content analyzed by fluorescence spectroscopy of the product before and after purification. The content of O and Fe impurities were reduced from 15.1% and 5.21% before purification to 5.5% and 0.667% after purification, respectively. This phenomenon showed that the catalyst Fe in the product was effectively removed. Due to the corrosion resistance of boron nitride nanotubes and the blocking effect of bamboo knots, a small amount of stainless-steel particles are wrapped at the ends of the nanotubes, which are difficult to remove by acid cleaning. Therefore, some Fe still exists after pickling. The content of O after hydrolysis is still somewhat high. This phenomenon may be ascribed to the oxidation of the coral-like BN micro-/nanostructure at high temperature during the preparation process, since the furnace tube and reaction vessel are both made of Al_2_O_3_, which will release some oxygen. Of course, another possible source is a small amount of boron oxide that is not completely hydrolyzed. In addition, the content of Al, Cr, Si, Mn and other impurities is too low to be measured accurately, so they are not considered. Moreover, these impurities mainly originate from stainless-steel particles with very low content. Analysis of the fluorescence spectrum showed that the prepared coral-like BN micro-/nanostructures contained 78% boron nitride before purification and increased to 92% after purification by pickling and hydrolysis.

## 4. Conclusions

In conclusion, coral-like BN micro-/nanostructures were synthesized from boron oxide by a ball milling and annealing process. The obtained coral-like BN micro-/nanostructures consist of a bamboo-shaped nanotube stem and dense BN flakes growing outward vertically on the surface of the trunk. The formation mechanism of the coral-like BN micro-/nanostructures is a two-stage growth process. Bamboo-shaped BN nanotubes firstly grow from milled B_2_O_3_/Fe powder under flowing NH_3_ gas and then many nanoflakes grow outwards on the surface of the BN nanotubes. 

Due to the existence of dense nanoflakes on the surface of this structure, it often shows a high specific surface area and good hydrophobic properties, coupled with the high temperature resistance and corrosion resistance of boron nitride materials themselves. Such materials have broad application prospects in the fields of composite material additives, hydrogen storage, catalyst carriers, drug carriers, sewage treatment, oil–water separation, etc.

## 5. Patents

One patent from the work reported in this manuscript has been authorized in China.

## Figures and Tables

**Figure 1 nanomaterials-13-00753-f001:**
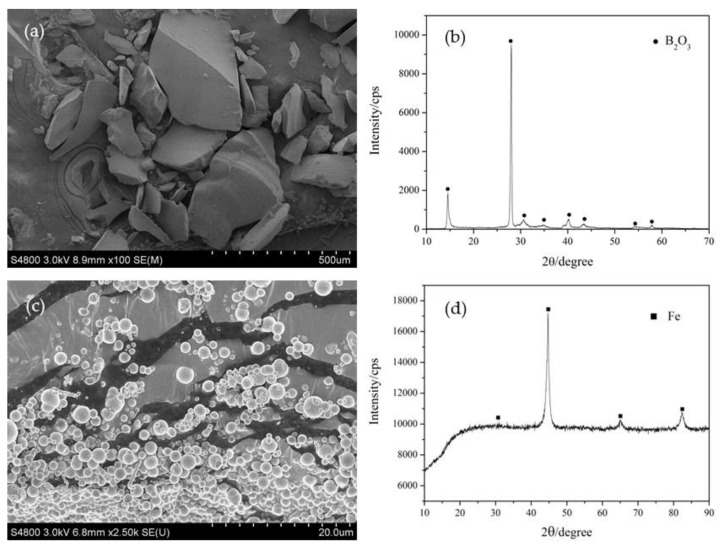
SEM images and XRD patterns of raw materials. (**a**) SEM image of B_2_O_3_, (**b**) XRD pattern of B_2_O_3_, (**c**) SEM image of Fe and (**d**) XRD pattern of Fe.

**Figure 2 nanomaterials-13-00753-f002:**
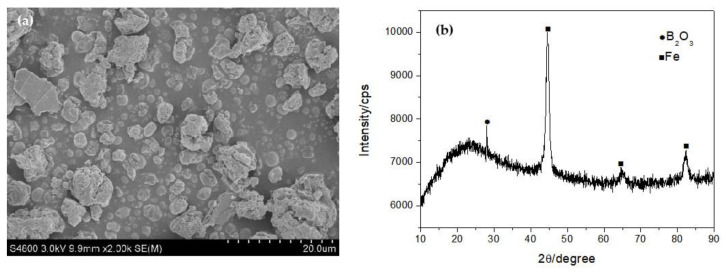
SEM image (**a**) and XRD patterns (**b**) of the product.

**Figure 3 nanomaterials-13-00753-f003:**
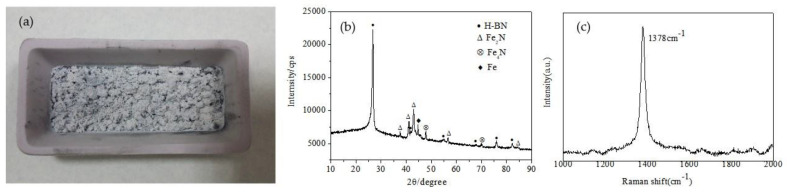
Optical image (**a**), XRD pattern (**b**) and Raman spectra (**c**) of the product.

**Figure 4 nanomaterials-13-00753-f004:**
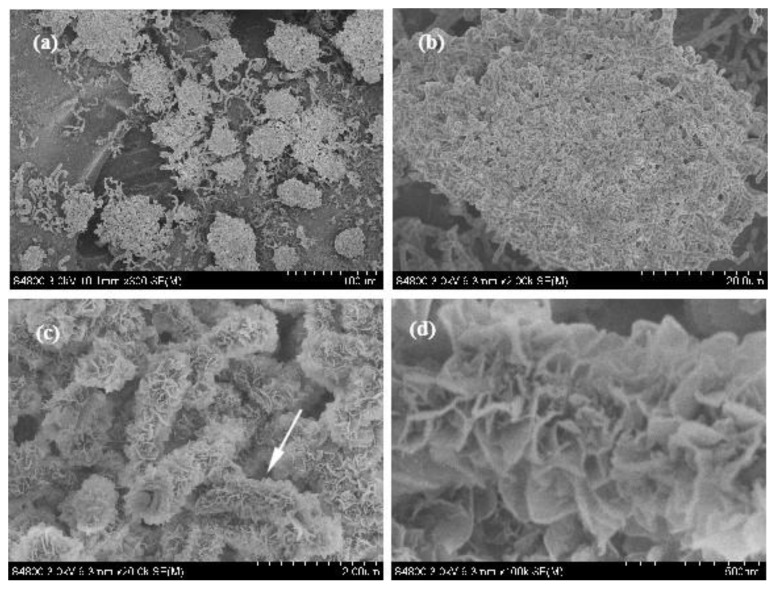
SEM image of the product with different magnification. (**a**) Low-magnification SEM image of the product; (**b**) SEM image of a single cluster; (**c**) enlarged SEM image of the cluster shown in (**b**); (**d**) Enlarged SEM image of single fibrous substances indicated by an arrow in (**c**).

**Figure 5 nanomaterials-13-00753-f005:**
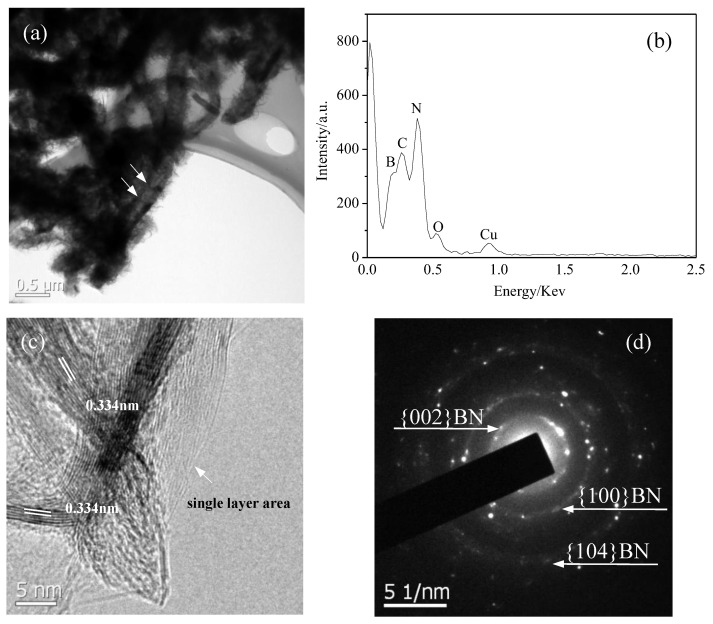
(**a**) TEM image of the product; (**b**) EDS spectrum of the product shown in (**a**); (**c**) HRTEM image of the flakes on the surface of the BN micro-/nanostructure; (**d**) SEAD pattern of the flakes shown in (**c**).

**Figure 6 nanomaterials-13-00753-f006:**
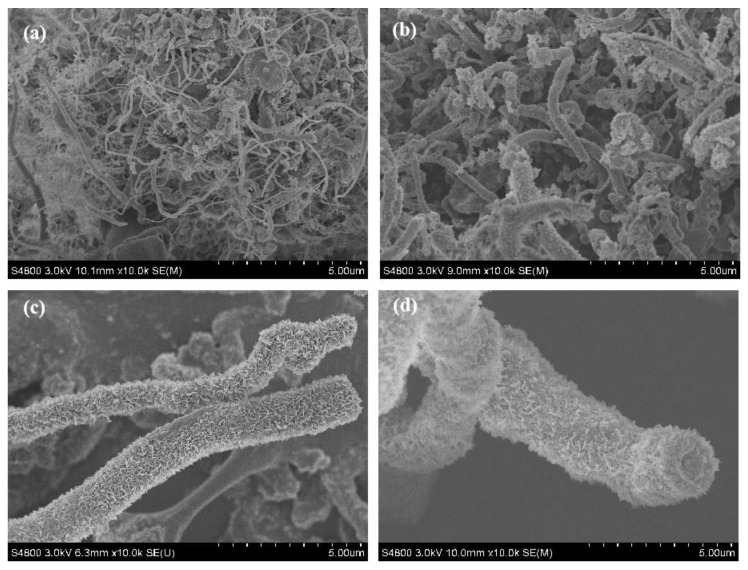
SEM images of the sample annealed at 1200 °C for different times: (**a**) 0.5 h, (**b**) 1 h, (**c**) 2 h, (**d**) 4 h.

**Figure 7 nanomaterials-13-00753-f007:**
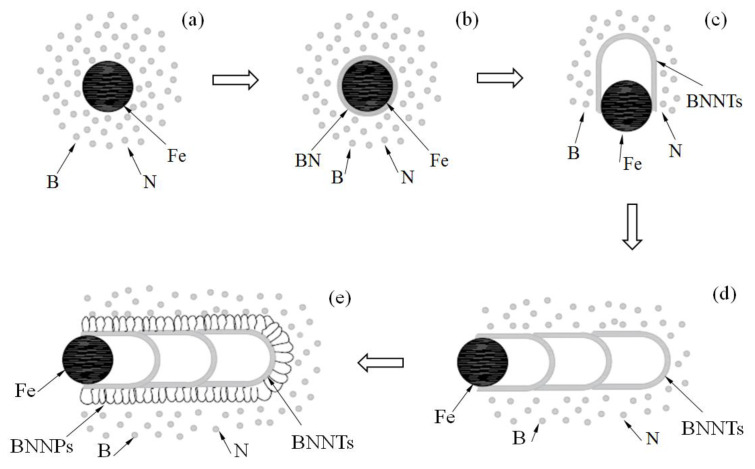
Growth process of the coral-like boron nitride micro-/nanostructure. (**a**) Melting of iron particles and adsorption of B, N atoms (**b**) Formation of BN crystal layer (**c**) Formation of single BN bamboo knot (**d**) Formation of bamboo-like BN nanotubes (**e**) Secondary growth of BN nanoflakes on the surface of BN nanotubes.

**Figure 8 nanomaterials-13-00753-f008:**
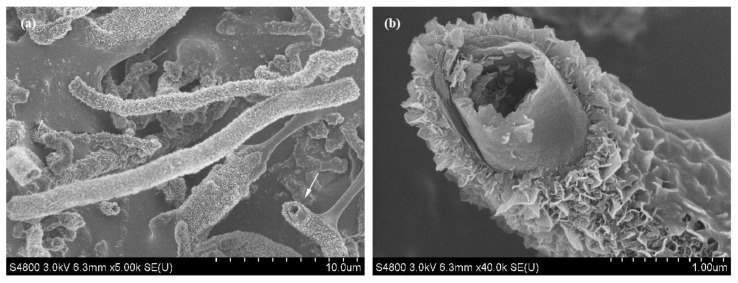
SEM image of the coral-like BN micro-/nanostructure. (**a**) Low-magnification SEM image of the coral-like BN micro-/nanostructure; (**b**) Partial enlarged SEM image of single coral-like BN micro-/nanostructure indicated by an arrow in (**a**).

**Figure 9 nanomaterials-13-00753-f009:**
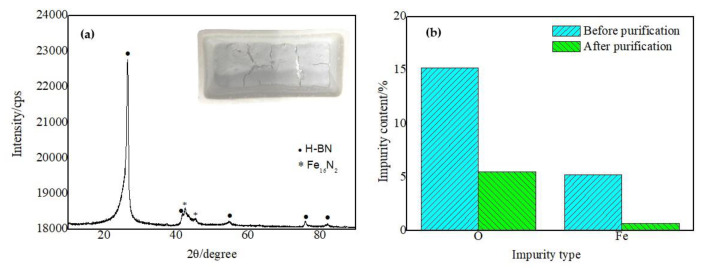
XRD pattern and optical image (inset) of the product after purification (**a**). Impurity content of products before and after purification (**b**).

## Data Availability

The data presented in this study are available on request from the corresponding authors.

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
