# Peer review of "A Simple Method for the Synthesis of a Coral-like Boron Nitride Micro-/Nanostructure Catalyzed by Fe"

_nanomaterials, 2023, doi:10.3390/nano13040753_

Round 1

Reviewer 1 Report

The ball milling and annealing techniques were utilized to prepare the coral-like h-BN nanomaterials. A series of characterization methods were used to confirm the structure. The formation mechanism was also investigated. This manuscript can be accepted after minor revision.

1.   Please provide the optical image of the synthetic h-BN power.

2.  From the XRD results, there still have some impurities in the h-BN powder, can these impurities be removed?

3.   Can the authors provide the Raman result?

4.    I think the authors should include more references on recent studies in h-BN, such as Nanotechnology 2019, 30 (24), 245706; Nanoscale 2022, 14 (11), 4204-4215; ACS applied materials & interfaces 2020, 12 (25), 28351-28359.

Reviewer 2 Report

The paper by Li et al reports a method for the synthesis of coral-like boron nitride structures. The synthesis procedure is quite interesting and well presented, however the authors investigated just the formation process of those structures. No further experimental characterization is reported. There are no analysis of the properties of the structures or their possible applications. The paper is only 7 pages long. This is not appropriate for a peer-reviewed publication. I suggest the authors to perform some more experimental activities, to strongly improve the introduction section and the state of the art, and to improve the bibliography of the paper as well and then to resubmit the manuscript.

Reviewer 3 Report

Li et al. report in this manuscript, with title “A simple method for synthesis of coral-like boron nitride micro-nano structure catalyzed by Fe“, on the preparation and characterization of a novel type of hexagonal boron nitride structure. The synthetic method is based on milled B2O3 powder doped with fine iron powder. The topic of the work is interesting, but there are some important issues which must be addressed:

1.- A very deep revision of the English must be performed throughout the manuscript.

2.- The introduction part is too short and must be elongated.

3.- The authors should give at least an estimation of the percentage of iron impurities found in the SEM image of figure 2. Besides, the “work distance” of all the SEM images should be indicated in their respective captions.

4.- Given that this manuscript is intended to be published in the Special Issue named “Ferro-/Piezoelectric Nanomaterials for Energy and Environmental Applications”, the conclusions section is not complete. It should give details regarding the possible applications of the reported material. So that, the conclusions section must be more developed.

5.- The authors should increase the number of references, which result to be a bit low for this research field.

Round 2

Reviewer 2 Report

The paper can be accepted in the present form

Reviewer 3 Report

I appreciate the efforts made to revise the manuscript and also the changes performed by the authors on it. These revisions make the work easier to be understood. Besides, I believe the results reported in the final version of the manuscript will also be appreciated by the readers of the journal Nanomaterials.